# Decoding Thalamic Glial Interplay in Multiple Sclerosis Through Proton Magnetic Resonance Spectroscopy and Positron Emission Tomography

**DOI:** 10.3390/ijms26178656

**Published:** 2025-09-05

**Authors:** Firat Kara, Nur Neyal, Michael G. Kamykowski, Christopher G. Schwarz, June Kendall-Thomas, Holly A. Morrison, Matthew L. Senjem, Scott A. Przybelski, Angela J. Fought, John D. Port, Dinesh K. Deelchand, Val J. Lowe, Gülin Öz, Kejal Kantarci, Orhun H. Kantarci, Burcu Zeydan

**Affiliations:** 1Department of Radiology, Mayo Clinic, Rochester, MN 55905, USA; neyal.nur@mayo.edu (N.N.); schwarz.christopher@mayo.edu (C.G.S.); kendallthomas.june15@mayo.edu (J.K.-T.); senjem.matthew1@mayo.edu (M.L.S.); port.john@mayo.edu (J.D.P.); vlowe@mayo.edu (V.J.L.); kantarci.kejal@mayo.edu (K.K.); 2Department of Information Technology, Mayo Clinic, Rochester, MN 55905, USA; kamykowski.michael@mayo.edu; 3Mayo Clinic Center for MS and Autoimmune Neurology, Rochester, MN 55905, USA; morrison.holly@mayo.edu; 4Department of Quantitative Health Sciences, Mayo Clinic, Rochester, MN 55905, USA; przybelski.scott@mayo.edu (S.A.P.); fought.angela@mayo.edu (A.J.F.); 5Center for Magnetic Resonance Research, Department of Radiology, University of Minnesota, Minneapolis, MN 55455, USA; deelc001@umn.edu (D.K.D.); gulin@cmrr.umn.edu (G.Ö.); 6Mayo Clinic Women’s Health Research Center, Rochester, MN 55905, USA; 7Department of Neurology, Mayo Clinic-Minnesota, Rochester, MN 55905, USA; kantarci.orhun@mayo.edu

**Keywords:** multiple sclerosis, thalamus, ^11^C-ER176 TSPO positron emission tomography, proton magnetic resonance spectroscopy, gliosis

## Abstract

The study assesses the relationship between thalamic proton-MR spectroscopy (^1^H-MRS) metabolites and thalamic ^11^C-ER176 translocator-protein positron emission tomography (TSPO-PET) standardized uptake value ratios (SUVR) to advance our understanding of thalamic involvement in multiple sclerosis (MS)-associated neurodegeneration and disability. In this prospective cross-sectional study, patients with MS (pwMS) and controls underwent 3T-MRI, ^1^H-MRS, and ^11^C-ER176-PET targeting the thalamus. MRI-derived thalamic volume was normalized by intracranial volume. ^1^H-MRS metabolites—N-acetylaspartate (NAA), glutamate (Glu), glutamine (Gln), total choline (tCho), and myo-inositol (mIns)—were normalized to total creatine (tCr). Clinical disability was evaluated using MS-specific tests of Expanded Disability Status Scale-EDSS and MS-functional composite-MSFC (including Paced Auditory Serial Addition Test-PASAT). Compared to controls (*n* = 30), pwMS (*n* = 21) exhibited smaller thalamic volume, higher thalamic ^1^H-MRS mIns/tCr (putative gliosis marker), and higher thalamic ^11^C-ER176-PET SUVR (glial density marker). In pwMS, higher thalamic mIns/tCr (r = −0.67) and tCho/tCr (r = −0.52) correlated with smaller thalamic volume. In pwMS, higher thalamic mIns/tCr correlated with higher thalamic ^11^C-ER176-PET SUVR (r = 0.48) and decreased cognitive function (PASAT, rho = −0.48). In controls, decreased thalamic NAA/tCr correlated with increased thalamic ^11^C-ER176-PET SUVR (r = −0.41). Thalamus, a core central nervous system relay, is affected early in MS disease course. Glial-mediated innate immune activation in the thalamus, evaluated by increased ^1^H-MRS mIns/tCr and ^11^C-ER176-PET SUVR, is associated with loss of thalamic volume and increased disability in pwMS. The multimodal imaging approach with ^1^H-MRS mIns/tCr and ^11^C-ER176-PET SUVR emerges as potential glial biomarkers, to better understand disease mechanisms and evaluate therapeutic interventions targeting glial activity in MS.

## 1. Introduction

Atrophy in the thalamus, a key central nervous system relay center, is among the earliest structural brain changes observed in patients with multiple sclerosis (MS) [1,2]. Given the pivotal role of the thalamus in sensory processing, motor coordination, and cognitive function, thalamic atrophy is clinically significant and strongly associated with disability in MS [1]. Recent research has focused on multimodal imaging of thalamic innate immune activation to elucidate its role in thalamic atrophy, associated with neurodegenerative features, and disability in MS.

Proton Magnetic Resonance Spectroscopy (^1^H-MRS) allows measurement of a range of brain metabolites, including myo-inositol (mIns), a putative marker of gliosis; total creatine (tCr), reflecting energy metabolism; glutamate (Glu) and glutamine (Gln), central to excitatory neurotransmission; N-acetylaspartate (NAA), a marker of neuronal integrity; and total choline (tCho), associated with membrane turnover [3]. ^1^H-MRS-mIns levels have been associated with gliosis, reflecting astrocytic and/or microglial activity, which are key components of the innate immune system [4,5]. Only a few studies have investigated mIns levels specifically in the thalamus of patients with MS (pwMS) [6,7,8,9], with two reporting increased mIns (mM) levels in pwMS [6,7].

Interpreting increased thalamic mIns levels is complex. While ^1^H-MRS mIns serves as a putative marker of gliosis, it also functions as an osmolyte [10]. Validating with independent imaging methods sensitive to glial activity or density is crucial to better understand the role of thalamic glial activation in pwMS and its link to neurodegeneration and disability. Therefore, in the current study, we coupled ^1^H-MRS metabolite alterations with 18 kDa translocator-protein (TSPO) positron emission tomography (PET), an advanced imaging method sensitive to innate immune activity. Coupling mIns levels with ^11^C-ER176 PET standardized uptake value ratio (SUVR) offers a unique opportunity to cross-validate mIns as a reliable in vivo marker of gliosis. Demonstrating a robust association between these two imaging modalities would provide converging evidence for the role of thalamic glial activation in MS and enhance the interpretability and utility of ^1^H-MRS in both research and clinical settings in MS.

A previous study using an older-generation TSPO radioligand (^11^C-PBR28) demonstrated that TSPO SUVR within the entire ^1^H-MRS voxel was not associated with ^1^H-MRS (mIns or mIns/tCr) in pwMS [11]. However, a subgroup with a high TSPO PET SUVR did show a positive relationship between the mIns/tCr ratio and TSPO SUVR [11]. Notably, that study neither examined the thalamic ^1^H-MRS metabolites nor investigated the correlations between mIns levels, thalamic atrophy, and MS disability, which are the specific foci of our study. To our knowledge, the association between thalamic ^1^H-MRS and thalamic TSPO PET SUVR using a third-generation radioligand ^11^C-ER176 to study thalamic glial activity/density, and its role in MS has not been investigated before.

The objective of this study was to compare thalamic ^1^H-MRS metabolite profiles between pwMS and controls, and to elucidate the correlation of thalamic ^1^H-MRS metabolites, particularly mIns/tCr, with thalamic ^11^C-ER176 PET SUVR, and thalamic volume in pwMS and controls. This molecular multimodal imaging approach aims to enhance our understanding of the interplay between elevated thalamic innate immune activation in MS-related neurodegeneration and disability, which may ultimately help uncover critical disease mechanisms and guide future therapeutic interventions in MS.

## 2. Results

### 2.1. Demographics

Twenty-one pwMS and 30 controls were prospectively enrolled in this study. The mean age (±SD) was 39.0 (±11.1) years for controls and 47.3 (±12.1) years for pwMS, with controls being significantly younger (*p* = 0.017). The sex distribution (pwMS women: 71% versus control women: 63%) did not differ between groups (*p* = 0.76). Among pwMS, 9 (43%) had no thalamic lesions, while 12 (57%) had thalamic lesions. Detailed cohort characteristics are summarized in Table 1 and Table 2.

### 2.2. Comparison of Imaging Parameters Between pwMS and Controls

No statistically significant interactions between groups and age were observed in any linear regression model (Appendix A), indicating that age did not influence the association between group (MS vs. control) and imaging parameters. Table 2 demonstrates a comparison of MRI, ^11^C-ER176 PET and ^1^H-MRS parameters between pwMS and controls after adjusting for age. Normalized thalamic volume (mean ± SD) was significantly smaller in pwMS (3.95 × 10^−3^ ± 0.50 × 10^−3^) than in controls (4.64 × 10^−3^ ± 0.46 × 10^−3^) (*p* < 0.001). On ^1^H-MRS, thalamic mIns/tCr ratios were higher in pwMS than in controls (*p* = 0.001). Additionally, thalamic ^11^C-ER176 PET SUVR was significantly higher in pwMS than in controls (*p* < 0.001).

### 2.3. Correlation of Thalamic ^1^H-MRS with Normalized Thalamic Volume and Thalamic ER176 PET SUVR

Table 3 demonstrates the age-adjusted correlation analysis between normalized thalamic volume and ^1^H-MRS metabolites in pwMS and controls. Smaller normalized thalamic volume correlated with higher thalamic mIns/tCr (*p* = 0.001) and higher thalamic tCho/tCr (*p* = 0.019) in pwMS (Table 3).

Table 4 demonstrates the age-adjusted correlation analysis between ^1^H-MRS metabolites and ^11^C-ER176 PET SUVR in pwMS and controls. In pwMS, higher thalamic mIns/tCr correlated with higher thalamic ^11^C-ER176 PET SUVR (*p* = 0.034) (Table 4). In controls, lower thalamic NAA/tCr correlated with higher thalamic ^11^C-ER176 PET SUVR (*p* = 0.031) (Table 4).

Figure 1 presents scatter plots showing the relationship between ^1^H-MRS metabolites, normalized thalamic volume and ^11^C-ER176 PET SUVR without age-adjustment.

### 2.4. Correlation of ^1^H-MRS Metabolites with MS-Specific Disability Metrics

In pwMS, increased thalamic mIns/tCr correlated with decreased Paced Auditory Serial Addition Test (PASAT) z-scores (*p* = 0.036) (Table 5). Neither the Expanded Disability Status Scale (EDSS) nor the Multiple Sclerosis Functional Composite (MSFC) demonstrated statistically significant correlations with any thalamic ^1^H-MRS metabolites.

The main associations, specifically the correlation between increased mIns/tCr and higher thalamic ^11^C-ER176 PET SUVR, as well as the correlation between increased mIns/tCr and smaller normalized thalamic volume, were consistent when using estimated mIns (millimolar, mM) as well (Appendix A). However, the results on estimated metabolite concentrations (mM) and their associations with normalized thalamic volume, thalamic ^11^C-ER176 PET SUVR, and MS-specific disability metrics are provided in Appendix A to keep the main manuscript focused and concise. The results of estimated metabolite concentrations and the potential bias related to these findings are discussed in the Appendix A.

## 3. Discussion

This study uniquely implemented advanced multimodal cellular and molecular neuroimaging tools to investigate the relationship between thalamic ^1^H-MRS metabolites, ^11^C-ER176 PET and normalized thalamic volume, thereby expanding our understanding of thalamic ^1^H-MRS metabolite associations with thalamic glial density, neurodegeneration and disability in MS. The results suggest that increased thalamic mIns/tCr ratios were associated with smaller normalized thalamic volume, greater thalamic ^11^C-ER176 PET SUVR, and ultimately decreased cognitive function in pwMS.

### 3.1. Smaller Normalized Thalamic Volume in pwMS

MRI-derived measures of brain volume have established thalamic volume loss as a promising structural biomarker in MS [12]. Consistent with prior research [13,14], we observed significantly smaller normalized thalamic volume in pwMS than in controls. Thalamic atrophy is a clinically relevant biomarker of neurodegeneration in MS, which may be exacerbated by the damage to connected white matter tracts passing through the thalamus [15,16]. Several proposed mechanisms underlie thalamic atrophy, including increased innate immune activity, gliosis, glutamate excitotoxicity, oxidative stress, and axonal damage, all of which may lead to neural injury or loss [17].

### 3.2. Increased Thalamic ^11^C-ER176 PET SUVR in PwMS

TSPO PET SUVR is commonly used as a marker of glial cell density and/or activity [18,19,20]. Our PET results, using a third-generation radioligand ^11^C-ER176, which improves specificity and eliminates genotype-based variability [21,22], confirmed earlier reports showing increased thalamic TSPO PET SUVR in pwMS compared to controls, as demonstrated with first- and second-generation ligands [13,23]. We recently reported that ^11^C-ER176 PET signal was higher in multiple gray matter regions, particularly in the deep gray matter regions, with the highest ^11^C-ER176 uptake displayed in the thalamus in pwMS compared to controls [24]. Moreover, we showed that higher ^11^C-ER176 uptake in the thalamus correlated with higher clinical disability as well as worse imaging biomarkers of neurodegeneration including thalamus atrophy [24]. Elevated thalamic ^11^C-ER176 PET SUVR in pwMS likely indicates ongoing innate immune responses to demyelination and neurodegeneration occurring diffusely throughout the brain, as well as specifically within the thalamus [13].

### 3.3. Increased Thalamic ^1^H-MRS [mIns/tCr] in PwMS

The current study results extend earlier ^1^H-MRS studies in pwMS by employing an advanced modified sLASER MRS protocol at 3T, which enhances spectral quality, voxel localization, and signal-to-noise ratio [25]. We observed an increased mIns/tCr ratio in the thalamus in pwMS compared to controls, consistent with prior research [6,7]. mIns/tCr is a putative marker of gliosis but also functions as an osmolyte. Moreover, TSPO expression is most often associated with microglial activation, but it can also reflect the activity of astrocytes [26,27]. Therefore, ^11^C-ER176 PET SUVR is a promising molecular biomarker for glial cell density and/or activity [18,19,20]. When our findings of increased thalamic mIns/tCr and ^11^C-ER176 PET SUVR were considered together, these results suggest increased glial cell density and/or activity in the thalamus in pwMS.

There were no significant differences in the thalamic tCho/tCr ratio between pwMS and controls, aligning with earlier reports on thalamic metabolite alterations [7,8,28]. While a previous study linked choline-containing compounds (tCho) to glial cells [29], our results suggest that mIns/tCr may serve as a more sensitive marker of gliosis in the thalamus.

In our study, NAA/tCr did not differ significantly between pwMS and controls, consistent with prior findings [7], though some studies report decreased thalamic NAA in pwMS [6,8,14,28]. Discrepancies may arise from differences in ^1^H-MRS protocols, and clinical characteristics. Additionally, MS-related innate immune activity and increased glial cell density may play roles in promoting not only injury but also repair and recovery in axons and neurons. NAA/tCr alterations in pwMS or mouse model of MS have been reported to be reversible, reflecting the dynamic interplay between injury and repair mechanisms in the central nervous system [30,31].

In the current study, metabolite levels were expressed relative to tCr (metabolite/tCr ratio) to reduce the variability due to scanner settings, voxel composition, and tissue relaxation time [32]. Brain tCr may vary in pwMS due to metabolic or glial changes, potentially confounding results [32]. However, we found no significant differences in partial volume-corrected tCr (mM) between pwMS and controls (Appendix A), aligning with previous studies.

### 3.4. Increased ^1^H-MRS (mIns/tCr) Correlates with Higher Thalamic ^11^C-ER176 PET SUVR and Smaller Normalized Thalamic Volume in pwMS

Attributing the increase in mIns/tCr to a single glial population is challenging due to the common interaction between astrocytes and microglia. Previous findings suggest that mIns/tCr is a putative marker of gliosis, with its association to specific cell types potentially varying depending on the underlying pathology [5,33,34]. The significant positive correlations between thalamic ^1^H-MRS (mIns/tCr) and thalamic ^11^C-ER176 PET SUVR in pwMS successfully reflect the complex interplay of glial populations in the context of innate immune activation, neuroinflammation and neurodegeneration in MS. To the best of our knowledge, this is the first study to describe the correlation between thalamic ^11^C-ER176 PET SUVR and ^1^H-MRS in pwMS simultaneously using two advanced cellular/molecular imaging modalities.

In the current study, decreased NAA/tCr correlated with increased ^11^C-ER176 PET SUVR in controls, potentially reflecting a direct link between glial activity and neuronal integrity. In pwMS, adaptive plasticity mechanisms may preserve NAA/tCr levels despite glial activation, which contributes to both axonal injury and repair processes, including remyelination [35]. This decoupling may be due to compensatory or adaptive plasticity mechanisms unique to MS pathology, where glial cells participate not only in injury but also in repair. Thus, the absence of a significant correlation in pwMS may reflect this more complex pathophysiological environment related to neural integrity, in contrast to the more stable and linear processes observed in controls. Furthermore, decreased normalized thalamic volume was associated with increased tCho/tCr ratios in pwMS but not in controls. These findings suggest that decreased normalized thalamic volume may be associated with increased membrane turnover and/or gliosis.

### 3.5. Increased Thalamic mIns/tCr on ^1^H-MRS Correlates with Decreased Cognitive Function in PwMS

To our knowledge, this study is the first to demonstrate a significant correlation between PASAT z-scores (a measure of cognitive function, particularly of information processing speed, working memory, and attention) and thalamic mIns/tCr in pwMS, highlighting the close connection between gliosis and cognitive impairment. While we focused on the thalamus as the critical relay for most white matter tracks in the central nervous system, a previous study reported a negative association between cognitive function (PASAT) and elevated mIns/tCr in the normal-appearing white matter [36]. These findings align with the broader understanding of MS pathology, where gliosis, glial cell density and innate immune activity are not confined to visible lesions but extend more diffusely to white matter and deep gray matter regions as key regions relevant to cognitive impairment in MS. Overall, these correlations underscore the critical role of thalamus in the disease’s pathophysiology and suggests that mIns/tCr could serve as a biomarker sensitive to cognitive worsening in MS if confirmed in longitudinal studies.

In the current study, none of the metabolite ratios statistically significantly correlated with EDSS or MSFC scores, possibly due to variability in pwMS characteristics, consistent with earlier studies [14]. While, the EDSS score showed a negative correlation with NAA/tCr supporting previous findings [37], this association did not reach statistical significance.

These results underscore the complex and multifactorial nature of pathological processes related to MS-related disability, where everything from overall lesional density, critical lesion location (e.g., spinal cord or thalamic lesions), amount of lesion level recovery, and overall and critically localized (e.g., thalamus) glial activity may play significant roles in driving clinical outcomes such as cognitive function.

### 3.6. Limitations and Future Directions

The cross-sectional design of the study precludes causal inferences. Longitudinal studies are needed to elucidate the temporal dynamics of neurodegeneration and gliosis in MS. The small sample size limits generalizability, and future research should include diverse cohorts across MS subtypes and stages. In this study, due to the limited sample size, our analysis adjusting the associations between ^1^H-MRS, other imaging variables, and clinical metrics was restricted to including age as the only covariate. Future studies with larger sample sizes are encouraged to investigate the impact of additional relevant covariates to further clarify these associations. ^1^H-MRS metabolite levels can be influenced by relaxation times of tissue water or metabolites (See Appendix A for more detailed discussion [28,38,39]). However, accounting for relaxation time weighting in ^1^H-MRS is challenging in pathological tissues due to time constraints during data acquisition. In our study, these effects were minimized by using a high repetition time (TR) and low echo time (TE) for ^1^H-MRS data acquisition. Furthermore, metabolite concentrations were normalized to tCr to reduce variability and account for potential differences in partial volume and relaxation time effects. However, concerns remain regarding potential disease-related changes in tCr levels, as these could influence its reliability as a reference. Importantly, estimated mIns (mM) showed the same directional associations as mIns/tCr with ^11^C-ER176 PET SUVR (Appendix A) and normalized thalamic volume (Appendix A). This consistency strengthens the conclusion that the observed associations between mIns/tCr with ^11^C-ER176 PET SUVR and normalized thalamic volume were independent of the use of tCr as a reference, adding robustness to our main findings. 

## 4. Methods

### 4.1. Study Design and Participants

This cross-sectional single-center study utilized data collected from prospective MRI and ^11^C-ER176 PET imaging (Mayo Clinic Advanced Imaging in MS—MCAIMS) scans obtained between October 2023 and June 2024, involving pwMS and controls. The diagnosis of MS was established using the MS diagnostic criteria [40,41]. The control dataset excluded individuals with any form of central nervous system involvement or disease. Only participants with both ^1^H-MRS and ^11^C-PET data were included in our study and analyses presented here were focused on thalamic ^1^H-MRS and its association with thalamic volume, ^11^C-ER176 PET and clinical assessments.

### 4.2. MRI Acquisition and Processing

All MRI and ^1^H-MRS were performed on a 3.0 Tesla scanner (Prisma, Siemens Healthcare, Erlangen, Germany, XA30 software and 64-channel head coil) at the Mayo Clinic Rochester, as previously described [42]. Three-dimensional (3D) T1-weighted magnetization-prepared rapid acquisition with gradient echo (MPRAGE) image volumes were obtained using the following parameters: TR/TE = 2530/3.65 ms, slice thickness = 1.0 mm, field-of-view 256 × 256 mm^2^, matrix size = 256 × 256, flip angle = 70°, spatial resolution = 1 × 1 × 1 mm^3^.

Thalamic volume (mL) was quantified using an in-house atlas-based region of interests on T1-weighted-3D-MPRAGE MRI and the Mayo Clinic Adult Lifespan Template [43]. The left and right thalamic volumes were summed to calculate total thalamic volume, which was then normalized to total intracranial volume (TIV).

3D high resolution FLAIR (spatial resolution = 1 mm^3^ ) was utilized for segmentation of lesions using a semi-automated segmentation algorithm as described earlier [44]. Lesion masks were inspected and edited by a radiologist. 

### 4.3. ^1^H-MRS

Single-voxel ^1^H-MRS data for each participant were acquired by a modified semi-localized adiabatic selective refocusing (sLASER) sequence with the following acquisition parameters (repetition time = 5000 ms; echo time = 30 ms; and 64 averages) as described previously [45,46,47,48,49,50]. The volume of interest (21 × 34 × 14 mm^3^) was manually positioned on a mid-sagittal T1 weighted image, including right and left thalamus regions by an experienced MRI technician (Figure 2). Thalamic voxels were consistently placed bilaterally and centered to ensure reproducibility across participants. Voxel positioning in thalamus was evaluated with mid-sagittal anterior landmark of mammillary body and posterior landmark of mesencephalic aqueduct to place the inferior edge of the voxel and superior landmark of fornix for the upper edge limit. The thalamic MRS voxel was composed of white matter (WM), gray matter (GM), and cerebrospinal fluid (CSF) partial volume fractions (Appendix A).

The processed sLASER spectra were quantified using LCModel 6.3 (Stephen Provencher, Oakville, ON, Canada). Metabolite levels were calculated in as (1) estimated metabolite concentrations (mM) using tissue water as an internal reference; and (2) as metabolite/tCr ratios by normalizing the estimated metabolite concentrations to total creatine (tCr) [47,51,52,53,54,55]. The acquisition and processing techniques of ^1^H-MRS were described earlier [46,47]. A detailed description is available in the Appendix A. Briefly, estimated metabolite concentration were corrected for partial volume effect using GM, WM, and CSF tissue partial volume fractions in thalamic ^1^H-MRS voxel (Appendix A) and T2 relaxation time of thalamus (96 ms) was taken into account in LCModel fitting for concentration calculations [8,56] (Appendix A). The tissue classification images were created using Mayo Clinic Adult Lifespan Template [57] (see Appendix A for more details).

For ^1^H-MRS data quality control, we followed the most recent ^1^H-MRS expert consensus recommendations [52]. Spectra were evaluated for lipid contaminations and insufficient water suppression, which may lead to sidebands and unwanted coherences. In addition, LCModel residuals, and baseline distortions were evaluated. Spectra were excluded when these artifacts hampered the quantification of the metabolites of interest. For quantitative quality control, we assessed the Cramér-Rao Lower Bounds (CRLB) of metabolites and signal-to-noise (SNR) (i.e., the ratio of the maximum in the spectrum minus baseline over the window to twice root mean square error residual) ratio of a single spectrum, and the full width at the half maximum amplitude (FWHM) of the water-unsuppressed signal and LCModel FWHM (i.e., a rough estimate of the linewidth in the in vivo spectrum). The group mean CRLB was checked separately for pwMS and controls. Concentrations with CRLB values of 999%, indicating failed fitting, were not included for mean CRLB calculations. The reliably quantified metabolites (CRLB < 20; FWHM ≤ 0.1 ppm [≤13 Hz] at 3T) were NAA, Glu, Gln, tCho, and mIns and tCr. Mean CRLBs (±standard deviation) were within acceptable limits in both groups (controls vs. pwMS: tCho: 4.23 ± 0.68 vs. 4.57 ± 1.03; NAA: 3.60 ± 0.72 vs. 3.71 ± 0.72; Glu: 6.10 ± 1.35 vs. 5.90 ± 1.41; Gln: 18.5 ± 7.9 vs. 16 ± 3.64; mIns: 4.80 ± 0.96 vs. 4.52 ± 1.03; tCr: 2.90 ± 0.48 vs. 3.10 ± 0.54). Spectral quality was adequate and comparable across groups, with NAA SNR >3 in all individual spectra prior to averaging (i.e., before summing the 64 transients) (group-averaged NAA SNR: 149.39 ± 23.78 in controls; 140.40 ± 28.11 in pwMS). The partial volume-corrected estimated concentration (mM) of thalamic tCr did not differ between pwMS and controls (Appendix A). Therefore, tCr was used as the denominator for metabolite ratio calculations. The partial volume-corrected estimated ^1^H-MRS concentrations are presented in the Appendix A.

### 4.4. ^11^C-ER176 PET

PET imaging was performed using a PET/CT scanner (Siemens Healthineers, Erlangen, Germany or GE Healthcare Milwaukee, WI, USA), operating in 3D mode using the third-generation radioligand, ^11^C-ER176 (IND# 149229) at the Mayo Clinic in Rochester. ^11^C-ER176 PET images were acquired at 60–90 min after injection of ^11^C-ER176 (518 MBq, 14 mCi; range, 370–666 MBq, 10–18 mCi) using four 5 min dynamic frames. Standardized uptake value ratios (SUVR) were normalized to the cerebellar crus [58]. In this cohort, an age-adjusted linear model showed no significant difference in cerebellar crus SUVR between pwMS and controls (*p* = 0.114), consistent with a prior report of no group differences in this region [24]. Based on these results, the cerebellar crus was selected as the reference region for SUVR normalization. Quantification was performed using an automated imaging pipeline [59]. No additional correction was applied for the low-affinity TSPO binding genotype, as this genotype is relatively rare [26,60,61,62,63]. Further details are provided in the Appendix A.

### 4.5. Clinical Assessments

For pwMS, clinical assessments included the EDSS, the MS Functional Composite (MSFC), and its three components: the 25-foot timed walk (25FTW), the 9-hole peg test (9HPT), and the PASAT [64,65,66,67]. The z-scores of PASAT, 9HPT (dominant hand) and 25FTW were averaged to calculate MSFC z-scores (see Appendix A) [65,68,69]. All clinical assessments were performed by a trained examiner blinded to the imaging results.

### 4.6. Statistical Analysis

Group differences were assessed using the Mann–Whitney U test (Wilcoxon Rank-Sum test) or linear regression. Fisher’s exact test was used to analyze categorical variables. Available data were used without imputation of missing values in the analysis.

Multiple linear regression models were fitted with and without interaction term to estimate whether age influenced the association between the group (pwMS and controls) and imaging variables (^11^C-ER176 PET SUVR, ^1^H-MRS and MRI). The age × group as an interaction term was not statistically significant and was excluded from the final models [70]. To evaluate group differences in imaging measures, multiple linear regression analyses were conducted with imaging metrics as the dependent variables, group (pwMS vs. controls) as the primary independent variable. Age was included as a covariate in all models. Linear regression models were used to evaluate whether the mean values of the dependent imaging variables differed between pwMS and controls after adjusting for age.

Correlations between imaging variables were assessed using partial Pearson correlation coefficients (r), while associations between MS clinical disability scores and imaging variables were evaluated using age-adjusted Spearman rank correlation coefficients (rho). Given pwMS group comprises 21 participants, only age, the key confounder for imaging studies, was used as a covariate for partial Spearman and partial Pearson correlations. Normality assumptions for linear regression and Pearson correlation were assessed using Q-Q plots and histograms of residuals. Linearity in linear regression was evaluated using residuals vs. fitted values plots, confirming the assumption of linearity and normality. Multicollinearity in the multiple linear regression models was tested using the variance inflation factor (VIF) for each independent variable [71]. Multicollinearity was defined as a VIF value ≥ 5. All independent variables had a VIF value < 5.

No corrections for Type 1 errors were applied. Although adjustments for multiple comparisons reduce the risk of type I errors (false positives), they increase the risk of type II errors (false negatives), potentially leading to missed significant findings [72]. However, we reported all exact *p*-values enabling readers to evaluate the robustness of our results. Statistical significance was determined by *p* < 0.05. Statistical analyses were conducted using the R software (version 4.4.1) within the RStudio environment (version 2024.09.1 + 394). 

## 5. Conclusions

This study highlights the unique and significant association of thalamic ^1^H-MRS (mIns/tCr) with thalamic ^11^C-ER176 PET SUVR as well as with thalamic atrophy and cognitive decline in pwMS. As reliable markers of glial-mediated innate immune activation, our ^1^H-MRS (mIns/tCr) and ^11^C-ER176 PET findings in pwMS emphasize the interplay between glial activity, glial density and neurodegeneration, and the potential contribution of innate immune activity to the clinical disability, including cognitive impairment. These findings further enhance our understanding of the cellular and molecular pathological mechanisms in the thalamus associated with MS, positioning ^1^H-MRS and ^11^C-ER176 PET detected glial activity and density as reliable and sensitive biomarkers of pathological alterations in the thalamus. Furthermore, thalamic ^1^H-MRS and ^11^C-ER176 PET hold promise as potential biomarkers in current and future MS therapeutic intervention trials targeting glial cells.

## Figures and Tables

**Figure 1 ijms-26-08656-f001:**
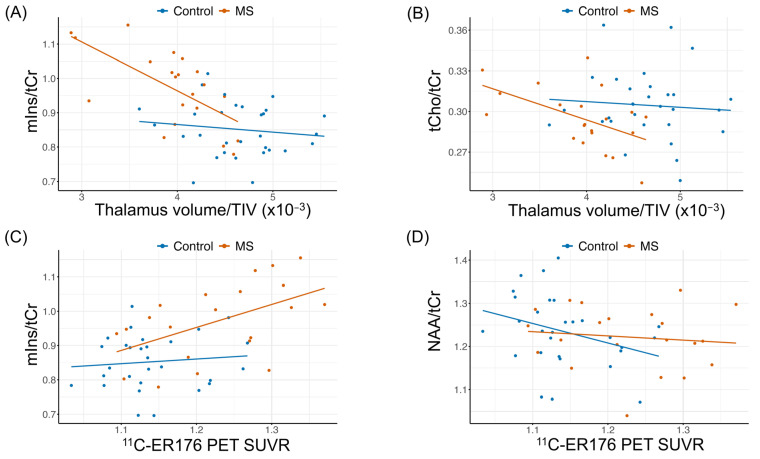
Scatter plots showing the relationships between (**A**) myo-inositol/total creatine (mIns/tCr) and normalized thalamic volume (thalamic volume/total intracranial volume, TIV); (**B**) total choline/tCr (tCho/tCr) and normalized thalamic volume; (**C**) mIns/tCr and ^11^C-ER176 PET standardized uptake value ratios (SUVR); and (**D**) N-acetylaspartate (NAA)/tCr and ^11^C-ER176 PET SUVR. The red dots = patients with multiple sclerosis (MS); blue dots = controls. The graphs display the original data points, focusing on original values rather than residuals. Trend lines show unadjusted linear regressions to visualize the overall direction of the relationships. Pearson correlation coefficients and *p*-values (r, *p*) of raw data from controls vs. pwMS are: (**A**): r = −0.13, *p* = 0.489 vs. r = −0.65, *p* = 0.002; (**B**): r = −0.08, *p* = 0.690 vs. r = −0.52, *p* = 0.016; (**C**): r = 0.10, *p* = 0.599 vs. r = 0.51, *p* = 0.017; (**D**): r = −0.32, *p* = 0.088 vs. r = −0.11, *p* = 0.62. Age-adjusted correlations and *p*-values are demonstrated in Table 3 and Table 4.

**Figure 2 ijms-26-08656-f002:**
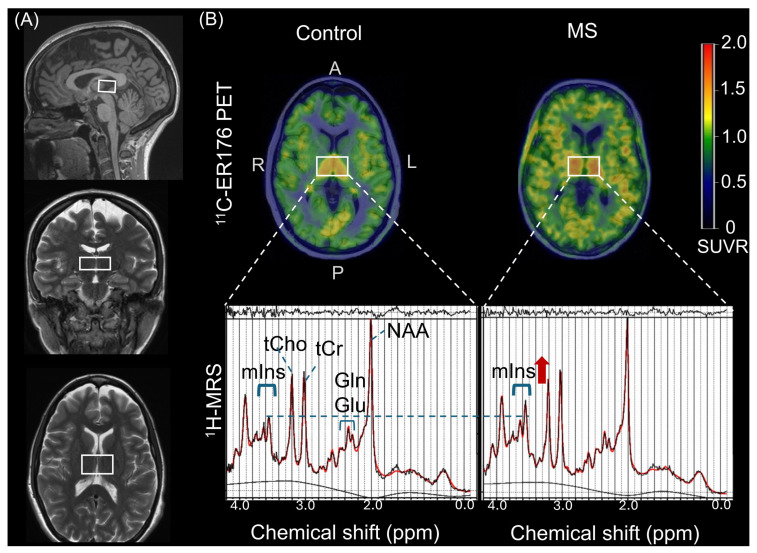
Sagittal, coronal, and axial T1–weighted magnetic resonance images with superimposed thalamus proton magnetic resonance spectroscopy volume of interest (21 × 34 × 14 mm^3^) (**A**). The figures show representative ^11^C-ER176 positron emission tomography (PET) scans from a healthy control without Multiple Sclerosis (MS) and a patient with MS (pwMS) (**B**). Below is representative single-voxel proton (^1^H) magnetic resonance (MR) spectrum acquired at 3T using the sLASER sequence from the same participants who underwent ^11^C-ER176 PET. The healthy control had an mIns/tCr ratio of 0.69 and a thalamus ^11^C-ER176 PET standardized uptake value ratio (SUVR) of 1.14. (**B**) The pwMS had an mIns/tCr ratio of 0.91 and a thalamus ^11^C-ER176 PET SUVR of 1.27. The color bar scale indicates SUVR. The thick red curve on the MR spectrum represents the LCModel fit to the data, while the thin curve beneath it shows the fitted baseline. The residuals (data minus fit) are displayed at the top of the spectrum. A white rectangular box on the PET image indicates the thalamus region. The blue dashed line marks the mIns level in a healthy control, while the red arrow highlights the elevated mIns signal in a pwMS. Abbreviations: NAA = N-acetylaspartate; Glu = glutamate; Gln = glutamine; tCr = phosphocreatine + creatine; tCho = phosphocholine + glycerophosphocholine; mIns = myo-inositol; ppm, parts per million (a unit to express the frequency of chemical shift of metabolites in the magnetic field); A: Anterior; P: Posterior; L: Left; R: Right.

**Table 1 ijms-26-08656-t001:** Characteristics of Patients with MS.

	MS (*N* = 21)
Sex	
F	15 (71%)
M	6 (29%)
MS phase	
Relapsing	14 (67%)
Progressive	7 (33%)
Disease duration	14.8 [6.3, 20.3]/14.9 (±9.9)
Age at MS onset (years)	31.9 [26.7, 34.6]/33.1 (±10.3)
Age at progressive MS onset (years)	46.0 [34.3, 51.5]/43.7 (±10.6)
Age at imaging (years)	48.0 [40.0, 54.0]/47.3 (±12.1)
DMT	
No	7 (33%)
Yes	14 (67%)
EDSS score	3 [1, 4]/2.81 (±2.12)
EDSS status	
Mild	10 (48%)
Moderate	7 (33%)
Severe	4 (19%)
MSFC z-score	−0.19 [−0.32, 0.31]/−0.04 (±0.56)
PASAT score	45.00 [38.75, 52.00]/44.25 (±9.66)
9HPT score (s)	22.35 [20.04, 26.98]/28.74 (±19.43)
25FTW (s)	4.90 [4.25, 6.40]/8.04 (±8.66)

Data are presented as median [IQR: 1st to 3rd quartile range]/mean (±SD), or count (%), depending on the variable type. Abbreviations: DMT, Disease-Modifying Therapy; EDSS, Expanded Disability Status Scale; EDSS status, mild (EDSS, 0.0–2.5), moderate (EDSS 3.0–5.5), and severe (EDSS ≥ 6.0); F, female; M, male; MS, Multiple Sclerosis; PASAT, Paced Auditory Serial Addition Test; 9HPT, 9-Hole Peg Test from dominant hand; 25FTW, Timed 25-Foot Walk Test; MSFC, Multiple Sclerosis Functional Composite (see Supplementary Methods MSFC Section). This table demonstrates all the available data, but there are some missing characteristics: disease duration, age at progressive MS onset, and MSFC, PASAT, 9HPT, 25FTW (*N* = 1).

**Table 2 ijms-26-08656-t002:** Characteristics and Imaging Variables of Controls and Patients with MS.

	Control (*N* = 30)	MS (*N* = 21)	*p*-Value (Age-Adjusted)
Sex			0.76 ^a^
F	19 (63%)	15 (71%)	
M	11 (37%)	6 (29%)	
Age at imaging (years)			**0.017 ^b^**
IQR (Q1, Q3)	38.5 [29.2, 47.5]	48.0 [40.0, 54.0]	
Mean (±SD)	39.0 (±11.1)	47.3 (±12.1)	
Thalamus volume (mL)			**1 × 10^−5^**
IQR (Q1, Q3)	6.70 [6.34, 7.17]	5.60 [5.35, 6.05]	
Mean (±SD)	6.72 (±0.72)	5.58 (±0.76)	
Total intracranial volume (mL)			0.52
IQR (Q1, Q3)	1463.15 [1348.13, 1558.50]	1383.83 [1339.77, 1492.54]	
Mean (±SD)	1454.57 (±144.92)	1413.36 (±110.85)	
Thalamus volume/TIV ×10^−3^			**1 × 10^−5^**
IQR (Q1, Q3)	4.64 [4.34, 4.92]	4.01 [3.86, 4.21]	
Mean (±SD)	4.64 (±0.46)	3.95 (±0.50)	
Thalamus ^11^C-ER176 PET SUVR			**2.7 × 10^−4^**
IQR (Q1, Q3)	1.13 [1.11, 1.17]	1.23 [1.15, 1.30]	
Mean (±SD)	1.14 (±0.06)	1.23 (±0.08)	
Thalamus ^1^H-MRS metabolites			
tCho/tCr			0.20
IQR (Q1, Q3)	0.3 [0.29, 0.32]	0.29 [0.28, 0.3]	
Mean (±SD)	0.30 (±0.03)	0.29 (±0.02)	
NAA/tCr			0.95
IQR (Q1, Q3)	1.24 [1.19, 1.30]	1.22 [1.19, 1.27]	
Mean (±SD)	1.24 (±0.09)	1.22 (±0.07)	
Glu/tCr			0.20
IQR (Q1, Q3)	1.14 [1.05, 1.24]	1.15 [1.11, 1.22]	
Mean (±SD)	1.14 (±0.14)	1.18 (±0.09)	
Gln/tCr			0.48
IQR (Q1, Q3)	0.41 [0.36, 0.49]	0.45 [0.42, 0.52]	
Mean (±SD)	0.42 (±0.10)	0.45 (±0.08)	
mIns/tCr			**5.5 × 10^−4^**
IQR (Q1, Q3)	0.84 [0.79, 0.91]	0.98 [0.91, 1.05]	
Mean (±SD)	0.85 (±0.08)	0.97 (±0.11)	

Data are shown as *n* (%) or median [IQR: 1st to 3rd quartile range, Q1–Q3] and mean (±SD). For continuous variables, all group comparisons (except for “age at imaging” and sex) were adjusted for “age” using linear regression model. Separate models were run for each MR spectroscopy, volume, or PET SUVR measure as the dependent variable, with “age” and “group” as independent variables. The *p*-values reported are the *p*-values of the estimates for the “group” variable, indicating whether the group differences are statistically significant after adjusting for “age at imaging”. ^a^ For sex, the *p*-value is from Fisher’s Exact Test. ^b^ *p*-value for “age at imaging” group comparison is from the Mann–Whitney U test. Abbreviations: ^11^C-ER176 PET SUVR, ER176 positron emission tomography standardized uptake value ratio; thalamic ^1^H-MRS metabolite ratio indicates metabolite/tCr ratio; tCr, total creatine; tCho, total choline; NAA, N-acetylaspartate; Glu, glutamate; Gln, glutamine; mIns, myo-inositol; TIV, total intracranial volume. This table demonstrates all the available data, but there are some missing characteristics: ^11^C-ER176 PET SUVR (*n* = 1). Bold *p*-values denote statistical significance (*p* < 0.05).

**Table 3 ijms-26-08656-t003:** Partial Pearson Correlation (age-adjusted) Between Thalamic ^1^H-MRS and Normalized Thalamic Volume.

	MS (*N* = 21)			Control (*N* = 30)		
^1^H-MRS Metabolites	r	95% CI	*p*-Value	r	95% CI	*p*-Value
tCho/tCr	−0.52	[−0.9, −0.14]	**0.019**	−0.08	[−0.45, 0.29]	0.678
NAA/tCr	0.32	[−0.11, 0.75]	0.175	−0.11	[−0.48, 0.26]	0.564
Glu/tCr	0.22	[−0.22, 0.66]	0.347	−0.10	[−0.47, 0.27]	0.617
Gln/tCr	−0.07	[−0.52, 0.38]	0.782	0.13	[−0.24, 0.5]	0.498
mIns/tCr	−0.67	[−1, −0.34]	**0.001**	−0.19	[−0.55, 0.17]	0.32

See Table 2 for abbreviations of metabolites: r, correlation coefficient; MS (number of participants = 21); Control (number of participants = 30); Bold *p*-values denote statistical significance (*p* < 0.05). Associations were assessed using age-adjusted Pearson correlation.

**Table 4 ijms-26-08656-t004:** Partial Pearson Correlation (age-adjusted) Between Thalamic ^1^H-MRS and ^11^C-ER176 PET SUVR.

	MS			Control		
^1^H-MRS Metabolites	r	95% CI	*p*-Value	r	95% CI	*p*-Value
tCho/tCr	0.24	[−0.2, 0.68]	0.319	−0.12	[−0.49, 0.25]	0.528
NAA/tCr	−0.04	[−0.49, 0.41]	0.876	−0.41	[−0.75, −0.07]	**0.031**
Glu/tCr	−0.13	[−0.58, 0.32]	0.578	−0.28	[−0.64, 0.08]	0.151
Gln/tCr	0.01	[−0.44, 0.46]	0.972	−0.19	[−0.56, 0.18]	0.327
mIns/tCr	0.48	[0.09, 0.87]	**0.034**	0.19	[−0.18, 0.56]	0.338

See Table 2 for abbreviations of metabolites: r, correlation coefficient; MS (number of participants = 21); Control (number of participants = 29); Bold *p*-values denote statistical significance (*p* < 0.05). Associations were assessed using age-adjusted Pearson correlation.

**Table 5 ijms-26-08656-t005:** Associations Between Thalamic ^1^H-MRS with Clinical Metrics in Patients with MS.

	MSFC z-Score (*N* = 21)			PASAT z-Score (*N* = 21)			EDSS Score (*N* = 21)		
^1^H-MRS Metabolites	Rho	95% CI	*p*-Value	Rho	95% CI	*p*-Value	Rho	95% CI	*p*-Value
tCho/tCr	0.26	[−0.19, 0.71]	0.275	0.13	[−0.33, 0.59]	0.604	0.13	[−0.32, 0.58]	0.580
NAA/tCr	0.15	[−0.31, 0.61]	0.546	0.03	[−0.43, 0.49]	0.907	−0.4	[−0.81, 0.01]	0.084
Glu/tCr	0.18	[−0.27, 0.63]	0.466	0.31	[−0.13, 0.75]	0.198	−0.12	[−0.57, 0.33]	0.627
Gln/tCr	0.12	[−0.34, 0.58]	0.628	0.23	[−0.22, 0.68]	0.340	0.04	[−0.41, 0.49]	0.868
mIns/tCr	−0.22	[−0.67, 0.23]	0.363	−0.48	[−0.89, −0.07]	**0.036**	0.13	[−0.32, 0.58]	0.588

Age-adjusted Partial Spearman correlation coefficient (rho) is presented with confidence interval [95% CI]. This non-parametric method is appropriate given the non-normal distribution of clinical disability measures. Bold *p*-values indicate statistically significant associations (*p* < 0.05). For abbreviations of the variables see Table 1 and Table 2.

## Data Availability

The data are not publicly available due to privacy or ethical restrictions. The data is stored in the Medidata repository. Qualified academic and industry researchers can request data, from the Mayo Clinic Advanced Imaging Research Center. Once a request is submitted, the committee sends the indicated principal investigator an email confirming that the request was received and giving a timeline for committee review.

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
