# Peer review of "Decoding Thalamic Glial Interplay in Multiple Sclerosis Through Proton Magnetic Resonance Spectroscopy and Positron Emission Tomography"

_ijms, 2025, doi:10.3390/ijms26178656_

Round 1

Reviewer 1 Report

Comments and Suggestions for Authors

• Section 3.3 text states that smaller normalized thalamic volume correlates with higher mIns/tCr (r = −0.67) and tCho/tCr (r = −0.52) in pwMS, and that higher mIns/tCr correlates positively with PET SUVR (r = 0.48).

•However, Table 3 lists a positive association between mIns/tCr and normalized thalamic volume in pwMS (r = +0.48), which contradicts both the text and Figure 2A.

•Table 4 lists a negative association between mIns/tCr and PET SUVR in pwMS (r = −0.67), contradicting Section 3.3 and Figure 2C (reported positive).

Please reconcile the directionality across text, tables, and figures

SUVR normalization: You use cerebellar crus as reference. Given possible cerebellar involvement in MS, please justify this reference region and show that crus uptake did not differ between groups.

Confirm voxel placement consistency (left vs right vs bilateral, and centering strategy), lesion status within the voxel, and report within-session SNR, FWHM, CRLB distributions in the main text (now in Supplement).

Author Response

Please enter "Please see the attachment."

Reviewer 2 Report

Comments and Suggestions for Authors

Comments to Authors :

This is an interesting multimodal study combining mainly PET-TSPO with 1H-MRS in MS patients versus controls, and also evaluating MRI volumetric and cognitive function parameters.

The following points should be addressed:

  1. In Figure 1 The MRS voxel-VOI includes both thalami and CSF; so was there contamination of thalamic tissue by CSF during MRS acquisition? Please explain futher in Methods.
  2. Relative 1H-MRS work investigating the metabolic profile of the thalamus in CIS patients that could be mentioned in the Introduction: https://pubmed.ncbi.nlm.nih.gov/37324507/ (DOI: 3892/etm.2023.12048)
  3. Where there any differences/correlations in the results in both modalities between the Relapsing MS group and the Progressive MS group of the study?
